# Exploring the Reciprocal Relationship between Depressive Symptoms and Cognitive Function among Chinese Older Adults

**DOI:** 10.3390/healthcare11212880

**Published:** 2023-11-01

**Authors:** Jiehua Lu, Yunchen Ruan

**Affiliations:** 1Department of Sociology, Peking University, No. 5 Yiheyuan Road, Haidian District, Beijing 100871, China; 2School of Humanities and Social Sciences, Fuzhou University, No. 2 Xueyuan Road, Fuzhou 350108, China

**Keywords:** depressive symptoms, cognitive function, reciprocal relationship, Chinese older adults, cross-lagged panel model

## Abstract

(1) Objectives: This study aims to investigate the bidirectional relationship between depressive symptoms and cognitive function among older adults in China, addressing a research gap in the context of developing nations. (2) Methods: A total of 3813 adults aged 60 and older participating in 2013, 2015, and 2018 waves of the China Health and Retirement Longitudinal Study (CHARLS) were included. A fixed-effects model and cross-lagged panel model (CLPM) was utilized. (3) Results: First, the results indicated that a significant negative correlation existed between depressive symptoms and cognitive function in older adults during the study period (β = −0.084, *p* < 0.001). Second, after controlling for unobserved confounding factors, the deterioration and improvement of depressive symptoms still significantly affected cognitive function (β = −0.055, *p* < 0.001). Third, using the cross-lagged panel model, we observed a reciprocal relationship between depressive symptoms (Dep) and cognitive function (Cog) among Chinese older adults (Dep2013 → Cog2015, β = −0.025, *p* < 0.01; Dep2015 → Cog2018, β = −0.028, *p* < 0.001; Cog2013 → Dep2015, β = −0.079, *p* < 0.01; Cog2015 → Dep2018, β = −0.085, *p* < 0.01). (4) Discussion: The reciprocal relationship between depressive symptoms and cognitive functioning in older adults emphasizes the need for integrated public health policies and clinical interventions, to develop comprehensive intervention strategies that simultaneously address depressive symptoms and cognitive decline.

## 1. Background

Preserving optimal cognitive functioning among older individuals is paramount for the realization of healthy aging. China, considering its dramatic demographic-aging trends, is facing significant issues concerning the cognitive health of its older adult populace. Recent estimates have indicated that, as of 2018, the number of individuals aged 60 and above afflicted with dementia in China stood at approximately 15.07 million. Within this demographic, approximately 9.83 million individuals were diagnosed with Alzheimer’s disease (AD) [1]. Furthermore, there were approximately 38.77 million cases of mild cognitive impairment (MCI) among individuals aged 60 and above, yielding a prevalence rate of approximately 15.5%. Cognitive health issues have a profound impact on the well-being of older adults in their later years. On the one hand, cognitive decline frequently co-occurs with physical health ailments and psychological distress in older adults, thereby elevating the risk of mortality. On the other hand, diminished cognitive function can curtail the societal and familial roles that older adults fulfill, diminishing their ability to navigate the challenges associated with aging. Consequently, it has become imperative to address the challenges posed by cognitive health risks in older adults, delay the decline in cognitive function among older adults in China, and mitigate the proportion of older adults affected by dementia.

### 1.1. The Impact of Late-Life Depressive Symptoms on Cognitive Function among Older Adults

In recent years, an increasing number of scholars have directed their attention toward examining the ramifications of late-life depressive symptoms on the cognitive function of older adults [2]. Late-life depressive symptoms stand out as a significant contributor to the global disease burden and represents a profound mental health concern that markedly influences quality of life in older adults. It manifests through physical symptoms, a waning interest in and enthusiasm for daily activities, and in severe cases, even suicidal tendencies [3]. Psychological and epidemiological investigations have established a connection between heightened symptoms of late-life depressive symptoms and decline in various cognitive domains, encompassing information-processing capacity, memory function, and executive function, among others [2,4].

Information-processing speed is the initial cognitive domain impacted by late-life depressive symptoms, rendering it a sensitive indicator of cognitive decline. Information-processing capacity pertains to an individual’s competency in receiving external stimuli, and replicating, processing, encoding, and retaining information gleaned from the external environment [5,6]. Depression is tightly intertwined with declines in memory function. Multiple studies have posited that older adults manifesting significant depressive symptoms often exhibit diminished activation within the working-memory-network region [7]. Furthermore, the inhibitory impact of depressive symptoms on episodic memory in older adults assumes greater prominence when contrasted with younger cohorts [8]. Late-life depressive symptoms manifest associations with impairments in executive function. Extensive research has underscored the fact that late-life depressive symptoms frequently co-occur with various cognitive deficits, prominently including executive function deficiencies [9]. Among the diverse components comprising executive function, inhibition function has emerged as the element most susceptible to the impact of late-life depressive symptoms [10]. Furthermore, late-life depressive symptoms not only exacerbate the decline of various cognitive functions, but also culminates in more-severe cognitive impairments; for example, late-life depressive symptoms constitute a pivotal risk factor for MCI [11]. Additionally, a profound nexus exists between late-life depressive symptoms and AD, with select studies proposing that late-life depressive symptoms exert a substantial influence on the likelihood of developing AD [12].

Late-life depressive symptoms expedite cognitive-function decline and exhibit a close association with conditions such as MCI and Alzheimer’s dementia [13]. Furthermore, certain studies have suggested the potential existence of a bidirectional causal link between late-life depressive symptoms and cognitive function [14,15,16]. However, to date, there remains insufficient empirical evidence to robustly substantiate this bidirectional causality.

### 1.2. The Bidirectional Causal Causality between Late-Life Depressive Symptoms and Cognitive Function

Compared to the impact of late-life depressive symptoms on cognitive functioning, how cognitive function may influence late-life depressive symptoms has only been explored to a limited extent [6,17]. Scholars have investigated this relationship using longitudinal data from a survey tracking older adults in the United States [18]. Their findings revealed a noteworthy correlation between cognitive impairments in older adults and subsequent depressive symptoms. Similarly, a previous longitudinal study observed a significant association between cognitive impairments—including memory function—recorded during baseline surveys and the subsequent emergence of heightened depressive symptoms in older adults [19]. Research has indicated that older adults often perceive cognitive decline and impairment as akin to physical decline, viewing it as an irreversible weakening process which, in turn, exacerbates depressive symptoms. In terms of specific cognitive domains, some researchers have delved into the connection between these cognitive functions and late-life depressive symptoms [20]. Their study demonstrated that declines in information-processing speed significantly elevate late-life depressive symptoms in older adults.

In summary, although many scholars have discussed the relationship between late-life depressive symptoms and cognitive function, the existing literature has mostly focused on the correlation between the two [21,22] or the influence of the former on the latter [4], lacking sufficient examination of the bidirectional causal relationship between the level of late-life depressive symptoms and cognitive function in older adults.

Furthermore, the results of the few studies exploring bidirectional causal relationships are inconsistent [16,17], and few studies have attempted to effectively control for the omitted variable bias between the two, leading to difficulties in establishing causal identification. Some studies have found that late-life depressive symptoms can predict cognitive function, while others have come to the opposite conclusion. For instance, a study utilizing data from the population-based Longitudinal Aging Study Amsterdam indicated that depressive symptoms in older adults at baseline could predict subsequent cognitive function, but the reverse was not true [6]. However, a longitudinal study on older Hispanic adults in the United States indicated that a decline in cognitive abilities could predict depressive symptoms among older adults, but the converse was not substantiated [14].

### 1.3. Theoretical Perspectives and Hypothesis

This study aims to investigate the bidirectional relationship between depressive symptoms and cognitive function among Chinese older adults. However, discussing the bidirectional relationship between depressive symptoms and cognitive function is only part of the work examining the declining health in older adults, which features multi-dimensionality. We hope to preliminarily explore the theoretical framework for constructing a multi-dimensional pattern of declining health in older adults by introducing theoretical perspectives such as healthy lifestyle theory and stress process theory.

From the perspective of a healthy lifestyle, late-life depressive symptoms may affect cognitive function in older adults through their daily lifestyle choices [23]. Healthy lifestyle theory suggests that higher levels of depression hinder older adults from maintaining healthy lifestyles, leading to reduced sleep quality, decreased frequency of physical exercise and social engagement, disruption of adherence to medication schedules, and an increased likelihood of engaging in health-risk behaviors such as alcohol abuse [24]. For example, existing research has found that late-life depressive symptoms can affect sleep quality in older adults [25], and sleep quality is highly correlated with cognitive function [26]. Based on these viewpoints, we propose the following hypothesis:

**Hypothesis** **1.**
*In terms of causality, depressive symptoms impact cognitive function in older adults, meaning that the higher the level of depression in older adults, the lower their cognitive function.*


Stress process theory suggests that the process of cognitive decline in older adults is similar to the process of disability and can lead to gradually restricted functional abilities, thus acting as a long-term chronic stressor [27]. A decline in cognitive function not only limits an individual’s functional abilities but also weakens their social-support network. The extent to which an older adult’s functional abilities are limited directly affects their difficulty in engaging in social interactions and fulfilling social roles. As a long-term stressor, cognitive decline can lead to continuous hardships in maintaining independent living, depleting psychological resources such as sense of control and self-esteem, which further exacerbates late-life depressive symptoms [28]. Based on these viewpoints, we propose the following hypothesis:

**Hypothesis** **2.**
*In terms of causality, cognitive function impacts late-life depressive symptoms, meaning that the lower the cognitive function in older adults, the higher the level of depression.*


## 2. Methods

### 2.1. Data Sources

This study uses data from three waves of the China Health and Retirement Longitudinal Study (CHARLS), including data from 2013, 2015, and 2018. CHARLS is a large-scale nationally representative household survey led by the National School of Development at Peking University. The survey covers 28 provinces (municipalities, autonomous regions) and 450 villages in China. The CHARLS questionnaire covers a wide range of individual and family information for middle-aged and older adults, including variables representing mental and physical health status, such as depressive symptoms, cognitive function, and self-rated health, as well as demographic variables like gender, age, marital status, and education level. These data provide support for exploring the causal relationship between depressive symptoms and cognitive function in Chinese older adults.

The baseline survey of CHARLS began in 2011, followed by tracking surveys in 2013, 2015, and 2018. The sample sizes for CHARLS 2013, 2015, and 2018 were 18,605, 21,095, and 19,816, respectively. Leveraging the advantages of CHARLS tracking data, this study combined data from 2013, 2015, and 2018 and processed it into panel data. Based on the research objectives, this study retained samples of older adults aged 60 and above, and removed samples with missing values in the variables, resulting in a final sample of 3813 individuals. The demographic and health characteristics of the participants are shown in Table 1.

### 2.2. Variable Definitions

The measurement of depressive symptoms in older adults was based on previous research [29,30]. We used the Center for Epidemiologic Studies Depression Scale (CES-D-10), which consists of 10 questions asking respondents about their feelings and behaviors within the past week. Eight questions are concerned with the frequency of various depression-related symptoms, while the other two questions are about the frequency of positive emotions in the past week. For each of the 10 questions, there are four possible answers: rarely or none of the time; some or a little of the time; occasionally or a moderate amount of the time; and most or all of the time. These answers are scored 0 (rarely or none of the time) to 3 (most or all of the time). After reversing the 2 positive-mood items, the scores for all 10 questions are summed. The depression scale score ranges from 0 to 30, with higher scores indicating higher levels of depression.

The dependent variable in this study is the cognitive function of the older adults. Cognitive function in older adults refers to the mental process of acquiring and processing information through senses, memory, reasoning, and decision making. Following previous research [31], we used two indicators—mental intactness and memory ability—to measure cognitive function in older adults. First, in the measurement of mental intactness using CHARLS questionnaire items, we mainly relied on nine questions related to the calculation, orientation, and drawing ability of respondents. For example, questions such as “What is 100 minus 7?”, “What day of the week is it today?”, and “Draw two overlapping pentagrams” are included. The number of correctly answered questions represents the mental intactness score, which ranges from 0 to 9. Second, in the measurement of memory ability, the CHARLS questionnaire provides a list of 10 common words and asks respondents to recall these words twice, once immediately and again after four minutes. The average of the correct answers to both recall sessions yields the memory ability score, ranging from 0 to 10. Overall, the sum of mental intactness and memory ability scores yields the cognitive function score for older adults, ranging from 0 to 19, with higher scores indicating stronger cognitive function. 

### 2.3. Data Analysis

The data analysis consisted of three main steps:

Step 1. Cross-sectional OLS analysis: In this step, we examined the results in existing research concerning the correlation between late-life depressive symptoms and cognitive function using an ordinary least squares (OLS) regression model. This helped to validate the relationships found in previous studies, which is that there is a significant correlation between depressive symptoms and cognitive function among Chinese older adults.

Step 2. Individual-level fixed-effects model analysis: The primary purpose of this step was to address the omitted variable bias. In the individual-level fixed-effects model, we introduce dummy variables for each individual to control for unobservable individual characteristics, which helps us to handle omitted variable bias in panel data. We used an individual-level fixed-effects model to control for unobserved individual-level factors that may affect both late-life depressive symptoms and cognitive function, such as genetic inheritance and personality traits. These individual-level factors were assumed to be time-invariant and not influenced by other variables in the study. By employing the fixed-effects model, we obtained more-reliable estimates of the causal relationship between late-life depressive symptoms and cognitive function.

Step 3. Cross-lagged panel model analysis: Considering the possibility of bidirectional causality, we used a CLPM to examine the causal relationship between late-life depressive symptoms and cognitive function over time. Cross-lagged panel model (CLPM) is a recognized powerful method for exploring dynamic relationships between variables. CLPM reflects the dynamic effects between variables through cross-lagged paths, that is, it constructs a path of effect (known as autoregressive effect) from the previous level of a variable to its current level, as well as a path of effect (known as cross-lagged effect) to the current level of another variable. The CLPM allowed us to explore the temporal ordering of the variables and determine whether the relationship between late-life depressive symptoms and cognitive function was consistent with a causal direction. This step provided further insight into the causal relationship between the two variables.

By employing these three steps in the analysis, we aimed to gain a comprehensive understanding of the causal relationship between late-life depressive symptoms and cognitive function in Chinese older adults, while addressing potential endogeneity issues.

### 2.4. Summary

Based on three waves of the China Health and Retirement Longitudinal Study (CHARLS), this study attempts to address potential endogeneity issues using individual-level fixed-effects and cross-lagged panel models (CLPM) to explore the reciprocal relationship between late-life depressive symptoms and cognitive function in older adults.

## 3. Results

### 3.1. Cross-Sectional OLS Analysis

This study aimed to investigate the bidirectional causal relationship between late-life depressive symptoms and cognitive function in Chinese older adults. Therefore, we replicated the conclusions of previous research and analyzed the correlation between the two variables as a basis for further examining their causal relationship. The correlation analysis serves as a bridge to eliminate the possibility of inconsistent research results in the subsequent discussion of causal relationships due to data and measurement issues.

Table 2 presents the cross-sectional OLS estimation results regarding the relationships between late-life depressive symptoms and cognitive function, mental intactness, and memory ability in the study year (2018). The results of the correlation analysis indicated a significant negative association between depressive symptoms and cognitive function, and mental intactness and memory ability in older adults.

In Model 1 of Table 2, with cognitive function as the dependent variable, the results showed that, regardless of whether control variables were included, there existed a significant negative association between late-life depressive symptoms and current cognitive function, indicating that an increase in depressive symptoms led to a significant decrease in current cognitive function (β = −0.084, *p* < 0.001).

The results for Model 2 demonstrated a significant negative correlation between late-life depressive symptoms and current mental intactness, regardless of the inclusion of control variables (β = −0.055, *p* < 0.001). Similarly, Model 3 supported a negative relationship between late-life depressive symptoms and current memory ability (β = −0.029, *p* < 0.001).

Based on the above results, we repeated the analysis using cross-sectional data from 2013 and 2015, and the consistent OLS regression results further supported the robustness of these relationships. Although the current correlation analysis could not test the research hypotheses directly, it provided a foundation for the next step of examining Hypotheses 1 and 2, specifically discussing the causal relationship between late-life depressive symptoms and cognitive function.

### 3.2. Fixed-Effects Model Analysis 

The first step towards causal inference in this study was addressing the issue of omitted variable bias, which was achieved using panel data to control for individual-level confounding factors that do not vary over time, regardless of whether these confounding factors are observable. By employing a fixed-effects model, the results in Table 3 reveal that, after controlling for time-invariant confounding factors, an increase in depressive symptoms among older adults led to a significant decline in cognitive function, mental intactness, and memory ability. This finding was consistent with the existing literature, as the significant impact of late-life depressive symptoms on cognitive function has been supported by numerous scholars [6].

Table 3 presents the fixed-effect estimates for the effects of late-life depressive symptoms on the cognitive function, mental intactness, and memory ability. An increase in depressive symptoms resulted in a significant decline in cognitive function for Chinese older adults, as well as markedly suppressing both mental intactness and memory ability.

Model 1 in Table 3 treats the cognitive function as the dependent variable. The results indicated that, for older adults, an increase in the severity of depressive symptoms was associated with a significant decline in cognitive function (β = −0.055, *p* < 0.001).

Similarly, Models 2 and 3 in Table 3 treat mental intactness and memory ability as the dependent variables, respectively. In both cases, the results indicated that, an increase in depressive symptoms among older adults significantly impaired their mental intactness (β = −0.038, *p* < 0.001) and memory ability (β = −0.017, *p* < 0.001).

### 3.3. Cross-Lagged Panel Model Analysis

Based on the estimation using the fixed-effects model as a foundation, we employed a CLPM to examine the bidirectional causal relationship between depressive symptoms and the cognitive function among older adults. Based on the statistical results of the CLPM, we observed a significant bidirectional causal relationship between depressive symptoms and the cognitive function. Specifically, current depressive symptoms significantly affected the cognitive function in the next period, while the current cognitive function could also predict depressive symptoms in the subsequent period. This mutual influence is also applicable to the relationship between depressive symptoms and mental intactness, as well as memory ability.

Figure 1 displays the results of the analysis regarding the relationship between depressive symptoms and the cognitive function. Both depressive symptoms and the cognitive function exhibited a strong autoregressive effect (Dep2013 → Dep2015, β = 0.572, *p* < 0.001; Dep2015 → Dep2018, β = 0.496, *p* < 0.001; Cog2013 → Cog2015, β = 0.480, *p* < 0.001; Cog2015 → Cog2018, β = 0.519, *p* < 0.001). Furthermore, while it remains unobservable, the co-variation of the two residuals (for instance, ε_1_ and ε_2_) indicates a co-movement between depressive symptoms and cognitive function, even after accounting for potential confounding variables. The bidirectional causal relationship of interest in this study is represented by two sets of cross-lagged coefficients.

Regarding the influence of depressive symptoms on cognitive function, both coefficients were negative and significant, indicating a significant negative impact of depressive symptoms on the cognitive function in the subsequent period (Dep2013 → Cog2015, β = −0.025, *p* < 0.01; Dep2015 → Cog2018, β = −0.028, *p* < 0.001). Concerning the effect of the cognitive function on depressive symptoms, both coefficients were also negative and significant, indicating that cognitive function among older adults can well-predict depressive symptoms in the subsequent period (Cog2013 → Dep2015, β = −0.079, *p* < 0.01; Cog2015 → Dep2018, β = −0.085, *p* < 0.01).

Figure 2 shows that depressive symptoms in 2013 had a significant negative impact on mental intactness in 2015 (β = −0.014, *p* < 0.05), and the depressive symptoms in 2015 also had a significant negative impact on mental intactness in 2018 (β = −0.017, *p* < 0.01). Additionally, mental intactness in 2013 significantly predicted depressive symptoms in 2015 (β = −0.095, *p* < 0.01), and mental intactness in 2015 also significantly predicted depressive symptoms in 2018 (β = −0.08, *p* < 0.05).

As can be seen from Figure 3, depressive symptoms in 2013 had a significant negative impact on memory ability in 2015 (β = −0.018, *p* < 0.001), while depressive symptoms in 2015 also had a significant negative impact on memory ability in 2018 (β = −0.021, *p* < 0.001). Furthermore, memory ability in 2013 significantly predicted depressive symptoms in 2015 (β = −0.105, *p* < 0.05), and memory ability in 2015 also significantly predicted depressive symptoms in 2018 (β = −0.156, *p* < 0.01).

By combining the results from Figure 1, Figure 2 and Figure 3, we found that, regardless of the cognitive function, mental intactness, or memory ability, the bidirectional causal patterns between depressive symptoms and the three cognitive indicators among Chinese older adults were very similar; that is, depressive symptoms among older adults had a significant predictive effect on the cognitive function, mental intactness, and memory ability in the subsequent period. Simultaneously, all three cognitive indicators had a significant impact on the level of depressive symptoms in the subsequent period.

## 4. Discussion

The main aims of the present study were (1) to retest the correlation between late-life depressive symptoms and cognitive function among older adults in existing research; (2) on the basis of addressing omitted variable bias, to examine the impact of changes in depressive symptoms on changes in cognitive function; (3) and to explore the bidirectional relationship between late-life depressive symptoms and cognitive function among older adults in China. 

The results showed that (1) a significant negative correlation existed between depressive symptoms and cognitive function among Chinese older adults; (2) after controlling for unobserved confounding factors, the deterioration and improvement of depressive symptoms still significantly affected cognitive function; (3) and a reciprocal relationship existed between depressive symptoms and cognitive function among Chinese older adults. Hence, the findings suggest that depressive symptoms influence cognitive function among older adults, and vice versa. The Hypotheses 1 and 2 are confirmed.

### 4.1. Association between Depression and Cognitive Function of Older Adults

The correlation analysis based on cross-sectional data demonstrated the existence of a significant negative correlation between the level of depression and the current cognitive function in older adults. This finding is consistent with previous research, as there is relative unanimity in the academic community regarding the negative relationship between depressive symptoms in Chinese older adults and various dimensions of cognitive function [32].

The results of this study not only support existing research on Chinese older adult populations but are also in line with findings from other countries [33]. For example, using cross-sectional data, some researchers observed a highly significant correlation between depression and declining cognitive function in older adults in India, another developing country [34]. A study based on developed countries suggested that, in the older adult population in the United Kingdom, depressive symptoms were significantly correlated with cognitive decline [35].

### 4.2. The Fixed Effect of Depressive Symptoms on the Cognitive Function of Older Adults

The results of the fixed-effects model analysis indicated that, after controlling for unobserved confounding factors, an increase in the level of depression in older adults led to a significant decline in cognitive function. At the level of the causal relationship, we found support for Hypothesis 1. Based on the results from Table 3, we believe that Hypothesis 1 is partially supported; meaning that, after controlling for some confounding factors, changes in depression significantly affected changes in the cognitive function of older adults.

The role of depressive symptoms in cognitive function has been extensively discussed by scholars [6]. Using longitudinal data, studies have pointed out the predictive effect of depressive symptoms on cognitive function in older adult women and found a significant impact of depression on various dimensions of cognitive function [36]. Some other scholars also utilized longitudinal data to examine the influence of baseline depression levels on subsequent cognitive function in older adults, and their results also supported a significant effect of depression on cognitive function [37]. The statistical results in Table 3 are consistent with the empirical findings of the aforementioned studies.

### 4.3. The Reciprocal Relationship between Depressive Symptoms and the Cognitive Function of Older Adults

Using the CLPM, this study revealed a bidirectional causal relationship between the level of depression in Chinese older adults and their cognitive functioning. Specifically, it was found that the level of depression significantly predicted cognitive functioning in the following period, and, conversely, cognitive functioning also had a significant impact on the subsequent level of depression. Based on the results obtained from the CLPM analysis, this study provides support for Hypotheses 1 and 2. This means that, at the causal level, it was evident that depression significantly affected the cognitive function of the older adults and, conversely, the cognitive function had a statistically validated role in influencing depressive symptoms.

To date, only a limited number of studies have explored the bidirectional causal relationship between depressive symptoms in older adults and their cognitive functioning [6,16]. One study examined the temporal associations between depressive symptoms and cognitive decline among Chinese older adults, and found that baseline depressive symptoms influenced the subsequent cognitive function [16]. However, over time, the decline in baseline cognitive function does not predict depressive symptoms. Our study results suggested that not only can depressive symptoms in older adults significantly predict cognitive function, but the reverse relationship also holds true.

### 4.4. Theoretical Perspectives: Exploration Multi-Dimensional Patterns of Declining Health in Older Adults

In this study, we explored the bidirectional causal relationship between late-life depressive symptoms and cognitive function, and the results are in line with the perspectives from health lifestyle theory, stress process theory, and pathophysiological processes theory. Our research findings first indicated that depressive symptoms in older adults lead to a decline in cognitive function. According to the health lifestyle hypothesis, late-life depressive symptoms significantly restrict healthy lifestyle choices, including reduced physical exercise and diminished sleep quality, ultimately exacerbating cognitive decline in older adults [38,39]. In addition to the health lifestyle hypothesis, the pathophysiological processes hypothesis suggests that long-term depression in older adults can induce toxic reactions related to glutamate or steroids, leading to pathophysiological processes such as atrophy of the frontal lobe and hippocampus that weaken cognitive function [40,41].

Furthermore, cognitive function also has a predictive role in depressive symptoms. Stress process theory posits that the decline in cognitive function in later life serves as a chronic stressor affecting the mental health of older individuals. Such a decline in cognitive function threatens their social role, performance, and engagement, thus weakening their sense of control and self-esteem at the psychological level [28]. This study discussed the bidirectional causal relationship between depressive symptoms and cognitive function in Chinese older adults, particularly highlighting the role of cognitive function in depressive symptoms from the perspective of stress process theory.

By examining the bidirectional relationship between late-life depressive symptoms and cognitive decline, this study provides a potential foundation for the integration of perspectives from health lifestyle theory, stress process theory, and pathophysiological processes theory, and offers a preliminary exploration into the construction of a theoretical framework for examining the multi-dimensional patterns of declining health in older adults [42].

### 4.5. Contribution, Limitations and Future Directions

Despite the consensus in the academic community on the correlation between depressive symptoms and cognitive function in older adults, there is an urgent need to extend the investigation from correlation analyses to establishing causal relationships between the two variables. Existing research on the bidirectional causal relationship between depressive symptoms and cognitive function is scarce and yields inconsistent conclusions [6,16]. In comparison to the existing literature, this study not only fills gaps in content but also advances the causal identification of the relationship between the two considered variables.

It is important to note that this study has two potential limitations. First, due to data constraints, the cognitive function-related measures discussed in this study only included measures of mental intactness and memory abilities. Future studies should further explore the predictive effects of late-life depressive symptoms on specific cognitive function, such as executive function and processing speed. Second, the analytical strategy of this study was to estimate the causal relationship in two steps, separately addressing omitted variable bias and bidirectional causality, without considering these two issues simultaneously. Therefore, in the research findings, the reciprocal relationship between depressive symptoms and cognitive function among older adults may be affected by omitted variable bias, including the existence of a common mechanism between depression and cognitive function or even the concurrence of multiple co-causes. Future research could address these issues by employing advanced statistical methods, such as maximum likelihood for cross-lagged panel models with fixed effects.

### 4.6. Implications

The study reveals a bidirectional causal relationship between depressive symptoms and cognitive functioning among Chinese older adults. This discovery has significant implications for public health policy and clinical practice. Firstly, the understanding of this reciprocal relationship deepens our knowledge of the complex interplay between mental health and cognitive function among older adults. It suggests that these two aspects should not be treated independently, but rather, an integrated approach should be adopted. Secondly, this study calls for government intervention in the mental health and cognitive well-being of the elderly population. There is a pressing need to develop comprehensive intervention strategies that simultaneously address depressive symptoms and cognitive decline. These strategies could include mental health promotion, improved access to mental health services, cognitive training programs, and strengthening of social-support networks.

## 5. Conclusions

Examining the relationship between late-life depressive symptoms and cognitive function in Chinese older adults has significant clinical implications for improving cognitive health in the older adult population in China, thus delaying cognitive decline. In this study, we revealed a significant negative correlation between the severity of late-life depressive symptoms and current cognitive function. After controlling for unobserved confounding factors, an increase in the level of depression was found to lead to a significant decline in cognitive function among Chinese older adults. Furthermore, a bidirectional causal relationship was identified between the severity of late-life depressive symptoms and cognitive function, with the severity of depression significantly predicting subsequent cognitive function, while cognitive function also exerted a significant influence on subsequent depression levels.

## Figures and Tables

**Figure 1 healthcare-11-02880-f001:**
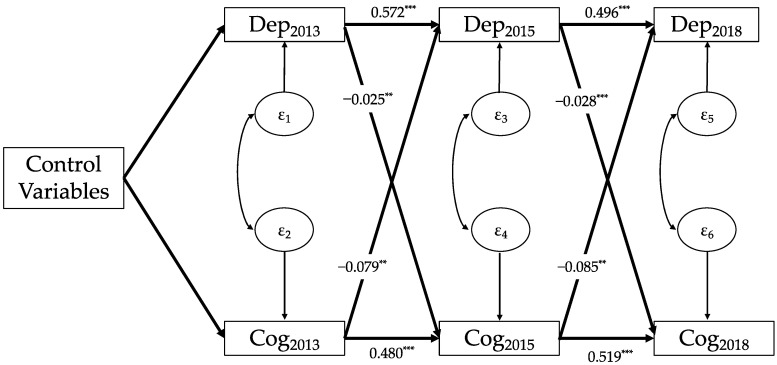
Cross-lagged panel model of depressive symptoms and cognitive function (N = 3813). Note: Data: CHARLS 2013, 2015, and 2018; Dep: depressive symptoms; Cog: cognitive function; subscript number of the variables mean survey wave; control variables: age, gender, region, educational level at baseline, marital status at baseline, self-reported health at baseline; ε_i_: residuals; and ** *p* < 0.01, *** *p* < 0.001.

**Figure 2 healthcare-11-02880-f002:**
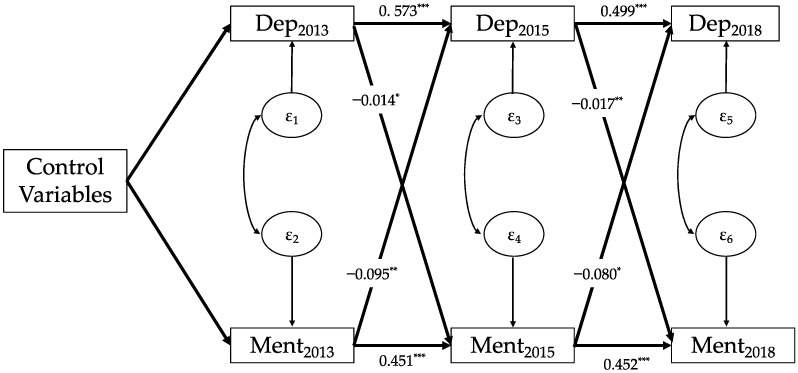
Cross-lagged panel model of depressive symptoms and mental intactness (N = 3813). Note: Data: CHARLS 2013, 2015, and 2018; Dep, depressive symptoms; Ment, mental intactness; subscript number of the variables mean survey wave; control variables: age, gender, region, educational level at baseline, marital status at baseline, self-reported health at baseline; ε_i_: residuals; and * *p* < 0.05, ** *p* < 0.01, *** *p* < 0.001.

**Figure 3 healthcare-11-02880-f003:**
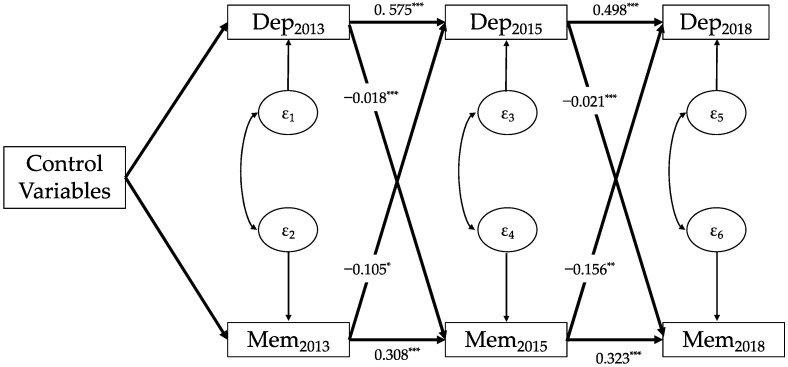
Cross-lagged panel model of depressive symptoms and memory ability (N = 3813). Note: Data: CHARLS 2013, 2015, and 2018; Dep, depressive symptoms; Mem, memory ability; subscript number of the variables mean survey wave; control variables: age, gender, region, educational level at baseline, marital status at baseline, self-reported health at baseline; ε_i_: residuals; and * *p* < 0.05, ** *p* < 0.01, *** *p* < 0.001.

**Table 1 healthcare-11-02880-t001:** Demographic and health characteristics of participants at baseline (N = 3813).

	N (%)	Mean (SD)
Cognitive function		9.59 (3.73)
Mental intactness		5.73 (2.67)
Memory ability		3.86 (1.76)
Depressive symptoms		7.77 (5.67)
Gender		
Male	2050 (53.76)	
Female	1763 (46.24)	
Age		66.11 (5.18)
Region		
Urban	983 (25.78)	
Rural	2830 (74.22)	
Marital status at baseline		
Currently married	3261 (85.52)	
Others	552 (14.48)	
Educational level at baseline		
No formal education	1797 (47.13)	
Primary school	1059 (27.77)	
Junior middle school and above	957 (25.10)	
Self-reported health at baseline		
Good	853 (22.37)	
Normal or poor	2960 (77.63)	

Note: Data: CHARLS 2013.

**Table 2 healthcare-11-02880-t002:** Ordinary least squares (OLS) regression model of cognitive function (N = 3813).

Variables	Cognitive Function (M1)		Mental Intactness (M2)		Memory Ability (M3)	
	Coefficient	SE	Coefficient	SE	Coefficient	SE
Depressive symptoms	−0.084 ***	0.009	−0.055 ***	0.007	−0.029 ***	0.005
Male	0.738 ***	0.108	0.798 ***	0.079	−0.059	0.057
Age	−0.092 ***	0.010	−0.050 ***	0.007	−0.042 ***	0.005
Urban	1.305 ***	0.128	0.750 ***	0.093	0.555 ***	0.067
Married	0.123	0.150	0.079	0.109	0.044	0.078
Education						
Primary school	2.287 ***	0.125	1.668 ***	0.091	0.618 ***	0.065
Junior middle school and above	3.303 ***	0.142	2.183 ***	0.104	1.119 ***	0.074
Self-reported health	0.109	0.125	0.048	0.091	0.061	0.065
R2	0.303		0.279		0.147	
Adj-R2	0.301		0.278		0.145	

Note: Data = CHARLS 2013; R2, R-squared; Adj-R2, Adjusted R-squared; SE, standard error; *** *p* < 0.001.

**Table 3 healthcare-11-02880-t003:** Fixed-effects models of cognitive function (N = 3813).

Variables	Cognitive Function (M1)		Mental Intactness (M2)		Memory Ability (M3)	
	Coefficient	SE	Coefficient	SE	Coefficient	SE
Depressive symptoms	−0.055 ***	0.009	−0.038 ***	0.007	−0.017 ***	0.005
Age	−0.113 ***	0.018	−0.067 ***	0.014	−0.049 ***	0.011
Married	0.393	0.269	0.162	0.203	0.231	0.161
Self-reported health	0.050	0.111	0.070	0.084	−0.020	0.067

Note: Data: CHARLS 2013 & 2015; SE, standard error; *** *p* < 0.001.

## Data Availability

Data are derived from public domain resources.

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
