# Peer review of "Exploring the Reciprocal Relationship between Depressive Symptoms and Cognitive Function among Chinese Older Adults"

_healthcare, 2023, doi:10.3390/healthcare11212880_

Round 1
Reviewer 1 Report
Comments and Suggestions for Authors
General Comment:
There is ample evidence that depression is a risk factor for subsequent cognitive decline and dementia, but evidence for a reciprocal relationship is less consistent. The study was therefore timely and pertinent.
Specific Comments:
1. Section 2.2. Variable Definitions, first paragraph, lines 159-160: The description of the scoring of the CES-D-10 is not correct. Actually, for each of the 10 questions there are three possible answers; these are scored 0 (rarely or none of the time) to 3 (most of the time). After reversing the two positive mood items, the 10 item scores are added up to give the total score.
2. Table 2: For readers who are less familiar with the type of analysis, it would be nice to see the definitions/explanations for R2 and Adj-R2 in the notes under the table.
3. Section 3.3. Cross-Lagged Analyses …, first paragraph, lines 265-269 (Existing research … the two considered variables.): It is tempting to mention this immediately after the description of the main result of the cross-lagged analyses, but it would be more appropriate to shift these two sentences to the discussion section.
4. Figures 1-3: For those who are less familiar with this type of analysis, it would be nice to find in the figure legends an explanation what ε means.
5. Discussion, fourth paragraph, 358-366: Apparently, something went wrong here with the references. After the sentence “To date, only a limited number of studies have explored the bidirectional causal relationship between depressive symptoms in older adults and their cognitive functioning“ reference is made to #6 and #36. However, in reference #36 no bidirectional causal relationship is reported. Then, a study in Chinese older adults is mentioned, but no reference is given. Reference #36 is given again for the statement that “over time, the decline in baseline cognitive function does not predict depressive symptoms.” I cannot find this in Reference #36. Please check.
Minor issue:
The relevant aspects of the findings are discussed appropriately. Please check, if the discussion can be condensed a bit without omitting important considerations.
Comments on the Quality of English LanguageAs far as I can tell, the English language quality appears quite good. Some minor editing might be useful.
Author Response
Dear Reviewer,
Thank you very much for your valuable comments and suggestions on our paper.
Please see the attachment.

Reviewer 2 Report
Comments and Suggestions for Authors
The research is methodologically sound and provides some interesting data on the controversial relationship between depression and cognitive functioning. However, it concludes a bidirectional causal relationship whose analyses need to be better described and substantiated. Here are some comments following my review:
· - Almost the entire introductory part is aimed at explaining how a non-causal correlation has been found between depressive symptoms and cognitive function and the desirability of exploring whether this relationship is causal or not. However, when establishing the hypotheses of the study, reference is made to theories that have not been explained or alluded to in the introductory part (healthy lifestyle and stress). On the other hand, in the discussion section, these theories are alluded to again, implying that the results of this research support and give support to them. In this sense, I ask myself the following question: is it really necessary to mention these theories to establish the hypotheses of the work or can the existence of significant correlations between depression and cognitive function and vice versa by themselves justify these hypotheses? If the answer is that the theories are necessary, they should be better stated in the introductory section.
· - Lines 178-179. “The demographic and health characteristics of the participants was shown in Table 1”. The reference to Table 1 should have been made earlier, in a section describing the sample or in the section "Data sources".
· - About point 3.3. Cross-Lagged Analyses of Depressive Symptoms on the Cognitive function of Older Adults. This section is complex to understand, but it is the most relevant of the study because it is where the bidirectional causal relationship between the variables studied is demonstrated. A better explanation of the analyses and results is needed, especially at this crucial point in the research.
· - Lines 367-390. The text implies that in some way these results support or sustain the aforementioned theories, however, without having concrete data regarding lifestyle, measures of stress level or concrete pathophysiological processes of the people in the sample, I find it difficult to conclude that the results obtained support these theories. In my opinion, it can be commented that the results are in line with what these theories tell us, but it cannot be assured that the results support these theories. On the other hand, other possible explanations such as the existence of a common substrate for cognitive functioning and depression rather than a causal relationship of either one over the other are not explored or even mentioned.
· - According to the above comment, the causality criterion chosen for the analyses seems to be precedence in time, but this is not the only criterion, nor in many cases the most important one. There are other explanations such as the existence of a common mechanism between depression and cognitive function or even the concurrence of multiple co-causes. These possibilities are not evaluated or included as possible limitations of the study.
· In the Discussion section, I miss some reference to the possible contributions of the results of the work, what could be the uses and consequences of these results? For example, establishment of vulnerable groups and prevention programs for these groups, effects on social and health policies, etc... Further reflection on the usefulness of the results is necessary.
Author Response

(The authors gave the same response as above.)

Reviewer 3 Report
Comments and Suggestions for Authors
The topic is fascinating, but a thorough restructuring of the content is necessary. I kindly request the esteemed author to revise the entire article with a deeper and more coherent perspective, taking into consideration the suggested changes. Afterward, please resubmit it for further review. I believe this feedback will be valuable in improving the quality of your work.
Best regards.
Abstract
- The objectives section is informative but could be made more concise. Consider rephrasing it for better clarity: "This study aims to investigate the bidirectional relationship between depressive symptoms and cognitive function among older adults in China, addressing a research gap in the context of developing nations."
- While this section provides a general overview of the methods used, it lacks specific details. You should include more information about the sample size, data collection methods, and statistical techniques used. This will help readers understand the study's rigor.
- Results should be more specific about the findings. Instead of using vague terms like "significant negative correlation," provide effect sizes or specific statistical results. For instance, mention the correlation coefficient and p-values. This will make the results more precise and easier to interpret.
- Also, include more context for the findings, such as the practical implications of the observed relationships between depressive symptoms and cognitive function.
- The discussion section is generally informative, but it could benefit from more in-depth exploration of the implications of the findings. Discuss the potential real-world applications and the broader significance of the study's results.
- Consider providing some suggestions for future research or practical interventions that might arise from this study. This can help readers see the study's contribution to the field.
Background
v While it provides substantial information, it is quite lengthy and could benefit from breaking it down into more concise and well-structured paragraphs. Group related information together to enhance readability and comprehension.
v Some sections are detailed but may contain more information than necessary for an introduction. Try to summarize information where possible to keep the reader engaged.
v The section starts with a strong focus on the impact of depressive symptoms on cognitive function but later introduces the bidirectional relationship. To improve clarity, consider introducing the bidirectional aspect earlier in the text.
v The hypotheses are important but could be presented more clearly. Consider placing them in a separate section or incorporating them into the text to highlight their significance.
v While discussing the relationship between late-life depressive symptoms and cognitive function, maintain a balanced view by mentioning any conflicting evidence or alternative theories. This will demonstrate a comprehensive understanding of the topic.
v Ensure consistent formatting and citation style for references throughout the section. This will make it easier for readers to locate the sources.
Methods
· While you've explained the data source and its relevance, you could include more details about the sample size at each wave (2013, 2015, and 2018) to provide readers with a better understanding of the dataset's scope and evolution over time.
· The definitions of depressive symptoms and cognitive function are well-explained. However, it would be helpful to provide some context or references for the scales used (CES-D-10) and the questions related to cognitive function. This would help readers understand the reliability and validity of the measurement tools.
· In Step 1, where you mention examining existing research concerning the correlation between late-life depressive symptoms and cognitive function, consider including the specific hypotheses or expectations you had before conducting this analysis. What were you trying to confirm or refute from previous studies?
· In Step 2, it would be helpful to explain the concept of an individual-level fixed-effects model to a non-expert audience briefly. This could be done in a sentence or two.
· In Step 3, where you discuss the cross-lagged panel model, it might be useful to explain what a cross-lagged panel model is in more detail and how it helps address bidirectional causality. This can help readers understand the methodology better.
· The Methods section is quite detailed, which is essential for transparency, but it might benefit from a more concise summary of the key points or findings from each step. This can serve as a quick reference for readers who want an overview of the methodology.
Results
- Throughout the section, maintain consistent reporting of significance levels (e.g., use *** for p<0.001 consistently).
Discussion
ü While the section is comprehensive, it is also quite lengthy. Consider breaking it into subsections for improved readability, with clear headings that guide the reader through the different aspects of the discussion.
ü Offer a concise summary at the beginning of the discussion, outlining the main findings to provide readers with a quick overview before delving into the details.
ü Emphasize the extent to which the results confirm or refute the hypotheses presented in the Introduction section. You mentioned that the study supports Hypotheses 1 and 2, which is valuable information, but it could be highlighted more explicitly.
ü The limitations are briefly mentioned at the end of the section. To provide a comprehensive assessment, consider expanding on each limitation's potential impact on the results and the study's overall validity. This would demonstrate your awareness of the study's limitations and the importance of addressing them in future research.
ü Expand on the practical implications of your findings. How can the results be applied to improve the well-being of older adults? This will make the study's significance more tangible to readers.
ü Discuss directions for future research. What specific areas or unanswered questions could follow-up studies explore? Suggest research avenues that may build upon your findings.
ü Make sure you maintain consistent reporting of significance levels (e.g., use *** for p<0.001 consistently).
Comments on the Quality of English Language
The topic is fascinating, but a thorough restructuring of the content is necessary. I kindly request the esteemed author to revise the entire article with a deeper and more coherent perspective, taking into consideration the suggested changes. Afterward, please resubmit it for further review. I believe this feedback will be valuable in improving the quality of your work.
Best regards.
Abstract
- The objectives section is informative but could be made more concise. Consider rephrasing it for better clarity: "This study aims to investigate the bidirectional relationship between depressive symptoms and cognitive function among older adults in China, addressing a research gap in the context of developing nations."
- While this section provides a general overview of the methods used, it lacks specific details. You should include more information about the sample size, data collection methods, and statistical techniques used. This will help readers understand the study's rigor.
- Results should be more specific about the findings. Instead of using vague terms like "significant negative correlation," provide effect sizes or specific statistical results. For instance, mention the correlation coefficient and p-values. This will make the results more precise and easier to interpret.
- Also, include more context for the findings, such as the practical implications of the observed relationships between depressive symptoms and cognitive function.
- The discussion section is generally informative, but it could benefit from more in-depth exploration of the implications of the findings. Discuss the potential real-world applications and the broader significance of the study's results.
- Consider providing some suggestions for future research or practical interventions that might arise from this study. This can help readers see the study's contribution to the field.
Background
v While it provides substantial information, it is quite lengthy and could benefit from breaking it down into more concise and well-structured paragraphs. Group related information together to enhance readability and comprehension.
v Some sections are detailed but may contain more information than necessary for an introduction. Try to summarize information where possible to keep the reader engaged.
v The section starts with a strong focus on the impact of depressive symptoms on cognitive function but later introduces the bidirectional relationship. To improve clarity, consider introducing the bidirectional aspect earlier in the text.
v The hypotheses are important but could be presented more clearly. Consider placing them in a separate section or incorporating them into the text to highlight their significance.
v While discussing the relationship between late-life depressive symptoms and cognitive function, maintain a balanced view by mentioning any conflicting evidence or alternative theories. This will demonstrate a comprehensive understanding of the topic.
v Ensure consistent formatting and citation style for references throughout the section. This will make it easier for readers to locate the sources.
Methods
· While you've explained the data source and its relevance, you could include more details about the sample size at each wave (2013, 2015, and 2018) to provide readers with a better understanding of the dataset's scope and evolution over time.
· The definitions of depressive symptoms and cognitive function are well-explained. However, it would be helpful to provide some context or references for the scales used (CES-D-10) and the questions related to cognitive function. This would help readers understand the reliability and validity of the measurement tools.
· In Step 1, where you mention examining existing research concerning the correlation between late-life depressive symptoms and cognitive function, consider including the specific hypotheses or expectations you had before conducting this analysis. What were you trying to confirm or refute from previous studies?
· In Step 2, it would be helpful to explain the concept of an individual-level fixed-effects model to a non-expert audience briefly. This could be done in a sentence or two.
· In Step 3, where you discuss the cross-lagged panel model, it might be useful to explain what a cross-lagged panel model is in more detail and how it helps address bidirectional causality. This can help readers understand the methodology better.
· The Methods section is quite detailed, which is essential for transparency, but it might benefit from a more concise summary of the key points or findings from each step. This can serve as a quick reference for readers who want an overview of the methodology.
Results
- Throughout the section, maintain consistent reporting of significance levels (e.g., use *** for p<0.001 consistently).
Discussion
ü While the section is comprehensive, it is also quite lengthy. Consider breaking it into subsections for improved readability, with clear headings that guide the reader through the different aspects of the discussion.
ü Offer a concise summary at the beginning of the discussion, outlining the main findings to provide readers with a quick overview before delving into the details.
ü Emphasize the extent to which the results confirm or refute the hypotheses presented in the Introduction section. You mentioned that the study supports Hypotheses 1 and 2, which is valuable information, but it could be highlighted more explicitly.
ü The limitations are briefly mentioned at the end of the section. To provide a comprehensive assessment, consider expanding on each limitation's potential impact on the results and the study's overall validity. This would demonstrate your awareness of the study's limitations and the importance of addressing them in future research.
ü Expand on the practical implications of your findings. How can the results be applied to improve the well-being of older adults? This will make the study's significance more tangible to readers.
ü Discuss directions for future research. What specific areas or unanswered questions could follow-up studies explore? Suggest research avenues that may build upon your findings.
ü Make sure you maintain consistent reporting of significance levels (e.g., use *** for p<0.001 consistently).
Author Response

(The authors gave the same response as above.)

Round 2
Reviewer 3 Report
Comments and Suggestions for Authors
The revisions to your paper have been diligently implemented, resulting in a fully corrected and now acceptable manuscript. We extend our gratitude for your dedicated efforts and collaborative approach in elevating the overall quality of your article.
We wish you continued success in your forthcoming research and writing endeavors.